# Evaluating Propagation Techniques for *Cannabis sativa* L. Cultivation: A Comparative Analysis of Soilless Methods and Aeroponic Parameters

**DOI:** 10.3390/plants13091256

**Published:** 2024-04-30

**Authors:** Matthew Weingarten, Neil Mattson, Heather Grab

**Affiliations:** School of Integrative Plant Science, Cornell University, Ithaca, NY 14853, USA; nsm47@cornell.edu (N.M.); hlc66@cornell.edu (H.G.)

**Keywords:** *Cannabis*, aeroponics, propagation, rooting, rockwool, horticultural (phenolic) foam, plant growth, nutrient concentration, soilless, sustainability, transplant, spray interval, cloning, vegetative propagation, horticulture

## Abstract

Given the rapid growth of the *Cannabis* industry, developing practices for producing young plants with limited genetic variation and efficient growth is crucial to achieving reliable and successful cultivation results. This study presents a multi-faceted experiment series analyzing propagation techniques for evaluating proficiency in the growth and development of *Cannabis* vegetative cuttings. This research encompasses various (1) soilless propagation methods including aeroponics, horticultural (phenolic) foam, and rockwool; (2) transplant timings; (3) aeroponic spray intervals; and (4) aeroponic reservoir nutrient concentrations, to elucidate their impact on rooting and growth parameters amongst two *Cannabis* cultivars. Aeroponics was as effective as, and in some cases more effective than, soilless propagation media for root development and plant growth. In aeroponic systems, continuous spray intervals, compared to intermittent, result in a better promotion of root initiation and plant growth. Moreover, raised nutrient concentrations in aeroponic propagation demonstrated greater rooting and growth. The effects of experimental treatment were dependent on the cultivar and sampling day. These findings offer valuable insights into how various propagation techniques and growth parameters can be tailored to enhance the production of vegetative cuttings. These results hold critical implications for cultivators intending to achieve premium harvests through efficient propagule methods and optimization strategies in the competitive *Cannabis* industry. Ultimately, our findings suggest that aeroponic propagation, compared to alternative soilless methods, is a rapid and efficient process for cultivating vegetative cuttings of *Cannabis* and offers sustainable advantages in resource conservation and preservation.

## 1. Introduction

*Cannabis* (*Cannabis sativa* L.) is an herbaceous annual plant, cultivated for millennia, serving purposes including industrial, food, medicinal, and recreational applications [1,2]. Recently, changes in legislation, a reduction in societal stigma, and advancements in newly permitted research have considerably increased its utilization and agricultural value. Optimizing cultivation practices ensures ideal potency, yield, and quality consistency, especially as the market landscape for human-consumed products becomes more competitive and subject to increasingly rigorous regulatory standards [3]. 

To meet these demands, a variety of propagation methods have been explored and adopted by *Cannabis* growers. These range from traditional practices, such as sowing seeds directly into the soil, to rapid and regenerative techniques such as vegetative propagation, in which stem cuttings from a stock plant are stimulated to root and produce genetically identical plants [4]. Another prevalent method is the use of tissue culture, a sophisticated approach that enables the generation of multiple plantlets from a small piece of plant tissue [5]. Each propagation method comes with its own set of advantages and challenges. While seeds offer genetic diversity, their germination rates, sex ratios, and genetic variation can be unpredictable [6]. Tissue culture, on the other hand, offers scalability and ensures disease-free propagules, but requires expensive facilities and equipment, as well as sterility and trained staff. Vegetative propagation through stem cuttings is the most common method for cannabinoid production, offering a low-cost approach compared to tissue culture and a more consistent outcome regarding genetics, gender, quality, and yield than seeds, but may risk the spread of pathogens. 

Variation in rooting and growth success during vegetative propagation has been observed in previous research with many plant species. Variability can be attributed to several factors such as choice of propagation method and genetics [7,8,9], in addition to stock plant age/health [10], cutting technique [11,12,13], VPD and temperature [12,13,14], hormone application [11], and others. There is an array of propagation systems used to produce vegetative propagules. In commercial usage, these typically include rockwool, horticultural foam, and aeroponics [15]. Rockwool, derived from molten basaltic rock, excels in maintaining a balance between water and air retention, in addition to promoting robust root growth [16,17,18]. Nonetheless, there are concerns about its sustainability through manufacturing, its single-use nature, and irrigation system maintenance, including the need for consistent moisture in propagation substances and humidity regulation [19]. Horticultural foam, crafted from petroleum-based phenolic foam, shares similar advantages and maintenance demands. Both of these options are single-use [15,19,20] and can often attract algae and/or mold over time. An alternative to these approaches is aeroponics, in which roots, enclosed inside a reservoir, are suspended and sprayed or misted consistently with a nutrient solution (fertigation). 

Several authors have claimed that aeroponic usage enhances nutrient absorption and oxygenation [21], as well as minimizing disease transmission risks, all while conserving water and other resources [22], aligning it with the global shift towards sustainable agriculture [18,23,24], though, to our knowledge, some of these claims have not been empirically evaluated. While the roots hang and are exposed to air, the system’s reliance on electricity poses a limitation [24]. Notably, Wimmerova et al. (2022) [25] provides a comprehensive comparison between aeroponic, hydroponic, and soil cultivation methods. Their study specifically highlights the enhanced growth and bioactive substance production in plants like *Coffea arabica* and *Senecio bicolor* when grown aeroponically, indicating that aeroponics not only addresses environmental concerns, but may also offer superior growth conditions compared to traditional techniques. Recent studies in *Cannabis* propagation using aeroponics have highlighted its potential benefits. Ferrini et al. (2021) [26] observed a considerable increase in root growth and a notable increase in secondary bioactive metabolites in aeroponic cultures compared to soil cultivation. In another study, Ferrini et al. (2022) [23] demonstrated that aeroponics significantly enhances the accumulation of bioactive constituents in the roots of *Cannabis*. There is, however, a lack of comparative studies evaluating aeroponics against other propagation systems or different aeroponic conditions, warranting further research in this area.

In order to meet the growing *Cannabis* industry’s need for sustainable and efficient cultivation methodologies, a series of studies was conducted to compare the efficacy of various soilless propagation systems, with a particular focus towards aeroponic optimization. Experiment 1 compares rooting success and propagule growth among aeroponics, horticultural foam, and rockwool. Experiment 2 assesses transplant growth at varied post-propagation intervals across the same systems. Experiment 3 examines the effect of different aeroponic pump timer intervals. Lastly, Experiment 4 analyzes the impact of reservoir nutrient strength and fertigation dilutions in aeroponics.

## 2. Results

### 2.1. Experiment 1–4

To provide a comprehensive overview of the effects of different propagation strategies on *Cannabis sativa* L. cuttings, a series of four experiments were performed. These experiments aimed to evaluate various growth metrics critical for identifying best practice strategies in plant propagation. The results of the experiments are organized by data point and experiment with a detailed outline (Table 1).

#### 2.1.1. Experiment 1: Propagation System

Root quality varied depending on the choice of propagation system (χ^2^ = 75.24, df = 2, *p* < 0.001), cultivar (χ^2^ = 43.85, df = 1, *p* < 0.001), and sampling day (χ^2^ = 181.74, df = 1, *p* < 0.001). The effect of the propagation system was dependent on the sampling day (χ^2^ = 10.21, df = 2, *p* = 0.006) and varied between sampling days and cultivars (χ^2^ = 7.95, df = 1, *p* < 0.005). On day 14, aeroponics had higher root scores in both cultivars when compared to horticultural foam, while only TJ’s CBD exhibited superior root scores in aeroponics compared to rockwool (Figure 1). By day 21, both Janet’s G and TJ’s CBD demonstrated enhanced root scores in aeroponics over rockwool, with TJ’s CBD also outperforming horticultural foam (Figure 1).

Propagule height varied based on the propagation system (χ^2^ = 28.12, df = 2, *p* < 0.001), cultivar (χ^2^ = 42.38, df = 1, *p* < 0.001), and sampling day (χ^2^ = 29.65, df = 1, *p* < 0.001). The effect of the propagation system depended on the sampling day (χ^2^ = 11.57, df = 2, *p* < 0.005), in addition to the cultivar (χ^2^ = 21.4, df = 2, *p* < 0.001). Janet’s G showed no difference in height among propagation systems at either 14 or 21 days. While for TJ’s CBD, there were no clear height differences on day 14, but aeroponics led to taller plants on day 21 compared to both horticultural foam and rockwool (Figure 2). 

The above-ground dry mass-to-stem diameter ratios varied across propagation systems (χ^2^ = 66.91, df = 2, *p* < 0.001) and cultivar (χ^2^ = 16.15, df = 1, *p* < 0.001). For Janet’s G, aeroponics consistently yielded higher masses compared to rockwool across both sampling days (Figure 3). For TJ’s CBD, aeroponics exhibited heavier masses than both horticultural foam and rockwool (Figure 3).

The below-ground dry mass to stem diameter was impacted by the propagation system (χ^2^ = 84.95, df = 2, *p* < 0.001) and cultivar (χ^2^ = 23.72, df = 1, *p* < 0.001). The propagation system was shown to have interactions between sampling day (χ^2^ = 37.7, df = 2, *p* < 0.001) and cultivar (χ^2^ = 10.81, df = 2, *p* < 0.005). On day 14, for TJ’s CBD, aeroponics had a greater below-ground dry mass compared to horticulture foam, but not rockwool, while Janet’s G showed no differences across propagation systems. By day 21, adjustments for media weight revealed that both Janet’s G and TJ’s CBD had consistently higher below-ground biomass in aeroponics, relative to other systems. This was after accounting for the average dry mass of the media—foam and rockwool—as these could not be separated from the roots (Figure 4).

#### 2.1.2. Experiment 2: Propagation System—Transplant

The effect of the propagation system on the growth of transplants was assessed 8, 10, 12, and 14 days after transplanting. Root quality score varied with cultivar (χ^2^ = 125.36, df = 21, *p* < 0.001) and propagation system (χ^2^ = 23.29, df = 2, *p* < 0.001). Interactions were shown between the propagation system and the cultivar (χ^2^ = 12.43, df = 2, *p* < 0.005), along with transplant day and cultivar (χ^2^ = 8.46, df = 3, *p* < 0.05). On day 10, horticulture foam outperformed aeroponics for Janet’s G. However, for TJ’s CBD, aeroponics outperformed rockwool (Figure 5). On day 12, for TJ’s CBD, both aeroponics and rockwool had higher root scores compared to horticulture foam, while Janet’s G showed no differences across propagation systems. On day 14, Janet’s G indicated higher root quality in aeroponics or horticulture foam over rockwool, while TJ’s CBD exhibited superior root quality compared to rockwool. 

The effect of height varied on the cultivar (χ^2^ = 146.1, df = 1, *p* < 0.001) and propagation system (χ^2^ = 18.62, df = 2, *p* < 0.001), along with the interaction between cultivar and propagation system (χ^2^ = 19.77, df = 2, *p* < 0.001). TJ’s CBD showed consistently taller plants in aeroponics compared to rockwool, but only outperformed horticultural foam on transplant days 10 and 14. Although, for Janet’s G, no clear differences among propagation systems or days were observed (Figure 6).

#### 2.1.3. Experiment 3: Aeroponics–Spray Interval

In Experiment 3, all cuttings were propagated in aeroponics and the effect of spray interval was evaluated. Root score varied by cultivar (χ^2^ = 111.83, df = 1, *p* < 0.001), spray time interval (χ^2^ = 32.71, df = 3, *p* < 0.001), and sampling day (χ^2^ = 286.65, df = 1, *p* < 0.001). The effect of the spray interval was dependent on sampling day (χ^2^ = 9.41, df = 3, *p* < 0.05), along with interactions between spray interval, sampling day, and cultivar (χ^2^ = 14.65, df = 1, *p* < 0.0001). On day 14, for both cultivars, constant spray showed an elevated root score over a 1 min on and 9 min off (1:9) time interval (Figure 7). In addition, on day 14 for TJ’s CBD, higher root scores were observed for continuous, 1:1, and 1:3, as compared to 1:9 (Figure 7). On sampling day 21, for TJ’s cultivar, continuous spray outperformed 1:9 (Figure 7).

The effect of spray interval on the height varied with spray interval (χ^2^ = 15.26, df = 3, *p* < 0.001), cultivar (χ^2^ = 105.38, df = 1, *p* < 0.001), and sampling day (χ^2^ = 129.34, df = 1, *p* < 0.001). Interactions were seen between sampling day and spray interval (χ^2^ = 27.77, df = 3, *p* < 0.001), along with sampling day and cultivar (χ^2^ = 31.13, df = 1, *p* < 0.001). Although, no clear differences in height were observed on day 14. On day 21, continuous spray displayed taller plants than 1:9 for Janet’s G. While, for TJ’s CBD, continuous, 1:1, and 1:3 had taller plants than 1:9 (Figure 8). 

The effect of the spray time interval on the above-ground dry mass-to-stem diameter depended on spray interval (χ^2^ = 22.49, df = 3, *p* < 0.001) and sampling day (χ^2^ = 21.14, df = 1, *p* < 0.001). Sampling day interactions were seen between spray time (χ^2^ = 18.16, df = 3, *p* < 0.0005) and with cultivar (χ^2^ = 26.21, df = 1, *p* < 0.001). On day 14, for Janet’s G, both continuous spray and 1:1 displayed larger dry masses than 1:9. On day 21, for TJ’s CBD, continuous spray performed better than all other spray intervals (Figure 9). 

The effect of the spray time interval on below-ground dry mass-to-stem diameter depended on spray interval (χ^2^ = 32.52, df = 3, *p* < 0.001), sampling day (χ^2^ = 158.06, df = 1, *p* < 0.001), and cultivar (χ^2^ = 108.82, df = 1, *p* < 0.001). The effect of spray intervals depended on sampling day (χ^2^ = 13.79, df = 3, *p* < 0.005) and cultivar (χ^2^ = 12.7, df = 3, *p* < 0.005), along with the sampling day’s interaction with cultivar (χ^2^ = 18.52, df = 1, *p* < 0.001). On day 14, for Janet’s G, the aeroponic’s 1:1 interval exhibited a higher dry root mass than the 1:3 interval (Figure 10), while TJ’s CBD demonstrated heavier results with aeroponics than both the 1:3 and 1:9 intervals. On sampling day 21, continuous spray resulted in a greater below-ground dry mass over all intermittent spray treatments for TJ’s CBD (Figure 10).

#### 2.1.4. Experiment 4: Aeroponics–Fertigation Dilution

In Experiment 4, the nutrient solution strength of the aeroponic fertigation water varied from 0.7 dS·m^−1^ to 1.4 dS·m^−1^. The effect of fertigation EC concentration on root quality score varied based on the concentration (χ^2^ = 21.17, df = 1, *p* < 0.001), sampling day (χ^2^ = 126.32, df = 1, *p* < 0.001), and cultivar (χ^2^ = 193.65, df = 1, *p* < 0.001). The effect of sampling day was dependent on cultivar (χ^2^ = 11.23, df = 1, *p* < 0.001). While Janet’s G had no clear differences among EC concentrations (Figure 11), TJ’s CBD, for both sampling days, demonstrated consistently higher root scores at an EC of 1.4 dS·m^−1^, compared to both 0.7 dS·m^−1^ (days 14 and 21) and an EC of 1.0 dS·m^−1^ (day 21) (Figure 11).

The effect of fertigation EC concentration on height varied with EC concentration (χ^2^ = 108.19, df = 2, *p* < 0.001), sampling day (χ^2^ = 189.38, df = 1, *p* < 0.001), and cultivar (χ^2^ = 184.39, df = 1, *p* < 0.001). The effect of EC concentration was dependent on sampling day (χ^2^ = 35.31, df = 2, *p* < 0.001) and cultivar (χ^2^ = 39.47, df = 2, *p* < 0.001), along with sampling day and cultivar (χ^2^ = 10.14, df = 2, *p* < 0.05). In addition, interactions were shown between sampling day and cultivar (χ^2^ = 53.15, df = 1, *p* < 0.001). Janet’s G exhibited a greater height at EC of 1.0 dS·m^−1^ or 1.4 dS·m^−1^ over 0.7 dS·m^−1^ on day 21 (Figure 12). TJ’s CBD exhibited a greater height at EC of 1.4 dS·m^−1^ vs. 0.7 dS·m^−1^ and 1.0 dS·m^−1^ on day 14 and, by day 21, height was greatest for an EC of 1.4 dS·m^−1^, followed by 1.0 dS·m^−1^ (Figure 12).

Above-ground dry mass-to-stem diameter varied based on the choice of EC concentration (χ^2^ = 33.68, df = 2, *p* < 0.001), sampling day (χ^2^ = 47.3, df = 1, *p* < 0.001), and cultivar (χ^2^ = 14.31, df = 1, *p* < 0.001). EC concentrations were shown to be dependent on cultivar (χ^2^ = 11.05, df = 2, *p* < 0.005) and sampling day (χ^2^ = 9.13, df = 2, *p* < 0.01), along with cultivar and sampling day (χ^2^ = 9.47, df = 2, *p* < 0.01). In addition, cultivar had interactions with sampling day (χ^2^ = 45.54, df = 1, *p* < 0.001). No significant effects of EC were noted for Janet’s G on either date. For TJ’s CBD, EC did not impact above-ground dry mass at day 14; but, at day 21, ECs of 1.0 dS·m^−1^ and 1.4 dS·m^−1^ had a greater mass than an EC of 0.7 dS·m^−1^ (Figure 13).

Below-ground dry mass-to-stem diameter varied depending on EC concentration (χ^2^ = 36.54, df = 2, *p* < 0.001), sampling day (χ^2^ = 145, df = 1, *p* < 0.001), and cultivar (χ^2^ = 166.76, df = 1, *p* < 0.001). The effect of EC concentration was dependent on sampling day (χ^2^ = 12.71, df = 2, *p* < 0.005) and cultivar (χ^2^ = 11.71, df = 2, *p* < 0.005), along with the sampling day-by-cultivar interaction (χ^2^ = 5.92, df = 2, *p* < 0.005), in addition to interactions between sampling day and cultivar (χ^2^ = 37.41, df = 1, *p* < 0.001). No clear trend was observed for Janet’s G and either date and TJ’s for day 14. For TJ’s CBD, at day 21, an EC of 1.4 dS·m^−1^ resulted in a superior below-ground mass than both ECs of 1.0 dS·m^−1^ and 0.7 dS·m^−1^, while an EC of 1.0 dS·m^−1^ had a greater mass than that of 0.7 dS·m^−1^ (Figure 14).

#### 2.1.5. Results Summary

In order to synthesize the data collected from our investigations into the efficacy of different treatments of *Cannabis sativa* L. cultivars, the findings from the four experiments have been comprehensively summarized (Table 2). The detailed data, separated by data point and experiment, provides a visual summary, allowing for an immediate comparison of results across different treatments and conditions. Further data and visuals are available, located in Appendix A.

## 3. Discussion

As global demand for *Cannabis* products continues to rise, cultivators are pressed to scale production while meeting evolving regulatory standards. Effective cultivation practices are necessary, especially the quality and uniformity of plant propagation, which can dictate the success of an entire crop. Among emerging solutions for higher value cannabinoid applications, soilless propagation practices stand out, offering the potential to produce *Cannabis* plants with limited genetic variation and efficient growth profiles. While commonly used materials like rockwool and petroleum-based (phenolic) horticultural foam are effective, they raise concerns due to their resource-intensive production processes and single-use nature [15,19]. As an alternative, this research evaluates aeroponic propagation and its impact on *Cannabis* growth and development under curated conditions. When comparing aeroponics to traditional soilless propagation methods, this study presents compelling evidence that aeroponics can yield equal or superior root and shoot development, promoting faster and healthier plant growth and transplant success. These experiments show that aeroponics offers a conservation-sensitive alternative to resource-intensive media. Furthermore, this investigation suggests the use of continuous spray and identifies optimized nutrient concentrations, promoting root and overall plant growth in aeroponics. 

### 3.1. Experiment 1: Propagation System

Experiment 1 demonstrated key distinctions between different soilless propagation systems and their impact on root development, plant height, above-ground dry mass, and root dry mass. Notably, aeroponic propagation performed as well as, and in some cases, better than, both horticultural foam and rockwool, in terms of promoting root score, plant height, and both above-ground and root dry mass. An observation for all four experiments is that treatments that lead to larger root-size generally also led to a larger shoot size. The observed greater variability in propagule height (Figure 2) at 21 days within aeroponic systems, compared to foam or rockwool, underscores the need for meticulous management to maintain growth uniformity, potentially through earlier transplant timing. This approach utilizes the already established root systems to achieve a more uniform growth across plants. The success of aeroponics can be attributed to its efficient nutrient and advantageous oxygen delivery, as highlighted in the study by Soffer and Burger (1988) [21], where increased dissolved oxygen concentrations significantly enhanced root formation and growth, while reducing dissolved oxygen concentration delayed root formation, decreased rooting percentages, reduced the number of roots per cutting, and shortened average root lengths in cuttings of *Ficus benjamina* L. and *Chrysanthemum* × *morifolium*. Further supporting our findings, the research by Yafuso et al. (2019) [17] on propagation methods revealed that maintaining the right balance of water and air within the media is crucial for healthy root development, as demonstrated through their analysis of peat, rockwool, and horticultural foam. Their findings on the high water content and limited air space at container capacity in these medias underline the importance of the aeroponic method’s superior air and moisture delivery system for *Cannabis* cultivation. The findings of this experiment align with previous studies, such as those by Ferrini et al. (2021) [26], who discerned that *Cannabis* plants cultivated in aeroponics for 8 weeks had a 13-fold higher root dry weight than their soil-grown counterparts. Our findings aim to contribute to sustainable cultivation methods in the *Cannabis* sector, not only for enhanced plant growth, but also for effective waste management and reduced environmental impact, as suggested by Robertson et al. (2023) [19].

### 3.2. Experiment 2: Propagation System–Transplant

Transplant success was influenced by the propagation system and the cultivar selection, with aeroponic propagation showing the greatest effect in enhancing transplant outcomes. Aeroponics performed as effective as, if not more than, both horticultural foam and rockwool, with higher root scores indicating its potential in enhancing transplant success and reduced transplant shock. The observed height variability (Figure 6) was demonstrated by taller aeroponic propagules on days 8 and 10, this aligns with the transplant success highlighted by propagation choice and cultivar selection, further emphasizing aeroponics’ capacity in increasing early transplant outcomes. The minimal impact of transplant timing in our results, which contrasts with the findings of Hinesley (1986) [8] on Fraser Fir seedlings, may be attributed to the unique advantages of aeroponics, as well as differences in plant species. Kumari and Kumar (2019) [24] emphasize aeroponics’ ability to optimize resource use and create a controlled environment for plant growth. This technology potentially mitigates the effects of transplant timing, offering a consistent and supportive growth environment, regardless of the time of transplant.

### 3.3. Experiment 3: Aeroponics–Spray Interval 

Exploring the impact of different aeroponic spray intervals on root and shoot development, this study builds upon the foundational insights of Weathers and Zobel (1992) [27], highlighting the critical role of hydration in the early stages of root development in aeroponic systems. Our findings reveal that continuous spraying or 1 min on and 1 min off timer intervals yield superior outcomes compared to longer intervals, emphasizing the importance of regular and consistent water spraying intervals for encouraging root initiation and plant growth. Complementing this, Tunio et al. (2021) and their subsequent work (2022) [28,29] investigated the effects of atomized nutrient solution droplet sizes and spraying intervals on aeroponic-grown butter-head lettuce. They observed that their shortest spray interval, 5 min on and 30 min off, using nozzles that produced smaller droplet sizes, significantly enhanced lateral root growth, biomass yield, and nutrient uptake, along with improvements in the root-to-shoot ratio, photosynthesis efficiency, and the nutritional quality of the plants. Research conducted by Tengli and Raju (2022) [30] on spray interval schedule and fertigation for aeroponic-grown potatoes demonstrates the crucial role of spray schedules in maximizing growth and yield. During their research, they observed that the misting cycle with the shortest interval time did not consistently yield the highest total yield. This suggests that factors beyond the shortest misting interval, such as overall misting duration and frequency, are pivotal for potato growth and yield in aeroponic systems and may vary depending on the plant’s requirements.

### 3.4. Experiment 4: Aeroponics–Fertigation Dilutions 

The observations indicated an EC of 1.4 dS·m^−1^ during *Cannabis* propagation yielded root and shoot growth results as good as, if not better than, those of 0.7–1.0 dS·m^−1^. This finding resonates with Raviv and Lieth (2007) [31], who articulated the variable nutrient needs of plants at different growth stages, emphasizing the importance of tailored nutrient management in soilless cultures. Supporting this, Caplan et al. (2017) [32] found that the highest yield and cannabinoid content in *Cannabis* were achieved with an organic fertilizer rate supplying approximately 389 mg N/L during the vegetative growth stage, highlighting the critical role of precise nutrient application in different cultivation mediums. Research on orchids by Wang (2000) [33] further supports this, demonstrating how specific nutrient adjustments can significantly impact plant development stages. Abdou et al. (2014) [34] also observed a positive response in Populus nigra L. seedlings to varied fertilization, which parallels our findings in *Cannabis*. Most notably, Wei et al. (2023) [35] demonstrated how different N, P, and K levels affected the growth and cannabinoid content of industrial *Cannabis* hemp, underscoring the impact of nutrient concentration on plant properties. Furthermore, research conducted by Tengli and Raju (2022) [30] on optimizing the nutrient formulation and spray schedule for aeroponically grown potatoes emphasizes the importance of tailored nutrient formulations and spray schedules in maximizing growth and yield. Their findings suggest that the optimal nutrient formulation and spray schedule may vary per species and growth stage, highlighting the need for species-specific optimization strategies in aeroponic agriculture. 

### 3.5. Variations and Future Directions

Genetic variation exists among cultivars [16]; in our experiments, TJ’s CBD had notably better survival rates, more rapid root initiation and overall growth. These cultivar-specific differences highlight the necessity of tailoring propagation strategies to better understand each cultivar’s requirements and possible potentials, considering their unique genetic makeup. Additionally, the influence of environmental conditions, such as temperature fluctuations within the greenhouse based on season, may have caused trial-specific variability. This was particularly noted in the Experiment 1 Propagation System and the Experiment 3 Spray Interval during Trial 1, however, each treatment demonstrated a similar pattern in subsequent trials. This trial occurred 12 April–02 May 2023, later in spring with warmer outdoor temperatures. The remainder of the trials and experiments benefited from climate control, having evaporative cooling on and the automatic deployment of retractable shade curtains. This highlights the impact of temperature regulation [36] on root growth, corresponding to previous research that noted a decrease in root meristematic speed and initiation when heat stress was experienced by the crop [37,38]. 

There are numerous sectors which can be studied to optimize *Cannabis* growth and productivity during the propagation and transplant establishment stages. Aeroponic and relevant propagation research potential exists in studying aspects such as broader ranges of EC concentrations and mixtures on nutrient absorption, reservoir water treatment amendments to further minimize disease risk, root architecture management, impacts of further environmental conditions, evaluating the environmental and economic impacts to reinforce aeroponics sustainable values, and exploring larger variations of propagation systems.

## 4. Materials and Methods

### 4.1. Greenhouse and Stock Plant Conditions

Stock plants of a CBD (cannabidiol) dominant cultivar, ‘TJ’s CBD’ (Stem Holdings, Boca Raton, FL, USA), and a CBG (cannabigerol) dominant cultivar, ‘Janet’s G’ (The Hemp Mine, Fair Play, SC, USA), along with the propagation trials were maintained at the Kenneth Post Greenhouses on Cornell University’s campus in Ithaca, New York. A 14 h photoperiod was provided with controlled supplemental canopy lighting from 400 W high-pressure sodium (HPS) lamps. Within the 14 h photoperiod, lights turned on when outdoor solar radiation was below 300 W·m^2^ and turned off when solar radiation was greater than 400 W·m^2^ for more than 10 min. In addition, low-intensity incandescent lights were turned on from 10 p.m. to 2 a.m. daily, to ensure plants perceived a short night length (maintaining vegetative growth stage). Temperatures averaged 26.0 ± 7.9 °C during the day and 18.3 ± 0.23 °C at night, with four days reaching above 32.0 °C for Trial 1 of Experiments 1 and 3, spanning 12 April through 02 May. Once evaporative cooling pads in the greenhouse were turned on, for the remainder of the trials, temperatures were less variable, averaging 26.1 ± 3.7 °C during the day and 20.3 ± 2.25 °C at night, through the remainder of the summer months. The closure of the retractable 50% shade cloth depended on solar intensity, as exposure to 10 min of direct sunlight at 600 W·m^2^ solar radiation triggered its closure. After 11 July, the shade cloth remained closed for reduced light intensity [39]. Stock plants were potted in 5 gallon pots containing a Lambert LM-111 All Purpose Mix (Lambert, Rivière-Ouelle (QC), CA) potting media. The stock plants were ~4 months old at experiment commencement. The plants were fertigated with Jack’s Professional LX All Purpose (JR PETERS Inc., #77990, Allentown, PA, USA) [21 N–2.18 P-16.5 K] (electrical conductivity, EC, 2.1 dS·m^−1^) on weekdays and with clear-water (0.5 EC) on weekends (add pH). Stock plants were scouted and treated weekly for pests and disease with ZeroTol 2.0 (BioSafe Systems, #70299-12, East Hartford, CT, USA), Cease (Bioworks, #264-1155-68539, Victor, NY, USA), Milstop (Bioworks, #68539-13, Victor, NY, USA), Ultra-Pure Oil (BASF, 69526-5-499, Mississauga (ON), CA), and Suffoil-X (Bioworks, #48813-1-68539, Victor, NY, USA).

### 4.2. Plant Culture and Treatment

For all experiments, cuttings were taken from apical branches of stock plants at a length of ~15–20 cm, having 2–3 fully expanded leaves and 3–5 nodes [11]. Each cutting was dipped in a Clonex 0.31% indole-3-butyric acid gel (Clonex, Growth Technology Ltd., Suffolk, UK) [40], before being placed at a 5 cm depth in each propagation system. 

### 4.3. Experiments

#### 4.3.1. Experiment 1: Propagation System

This experiment compared 64-site aeroponic propagation systems featuring macro droplet spray nozzles (Clone King, ck64, Albuquerque, NM, USA) to other soilless media treatments and was replicated twice. Although the aeroponic system in this research is a Clone King product, it shares common design concepts and elements found in many commercial aeroponic propagation systems [41]. Two popular soilless propagation medias—horticultural (phenolic) foam and rockwool—were selected, ROOTCUBES^®^ (Oasis Grower Solutions WEDGE^®^ strips, Kent, OH, USA) and rockwool cubes (AO Cubes, Grodan, Milton, ON, Canada).

The aeroponic unit was set to spray continuously, with a fertigated dilution of one-part nutrient solution and three-parts clear-water, resulting in four gallons per aeroponic unit (EC 1.0 dS·m^−1^). To maintain humidity in the rockwool and horticultural foam, 19.05 cm (7 ½ in) tall propagating domes with trays were used and plants were watered as needed with the same 1:3 nutrient dilution. Domes were kept closed for the first 6 days, then incrementally vented until day 14. Sets of 32 cuttings per cultivar were then placed in an aeroponic cloner, horticultural foam, and rockwool. The horticultural foam and rockwool were arranged randomly in 4 domed trays, each tray having 16 replicates of each cultivar across 2 replicates of each treatment. Aeroponics units contained 32 replicates of each cultivar, totaling 64 cuttings per unit. Aeroponic units were uncovered and exposed to greenhouse air movement. Each trial consisted of 192 cuttings total across all cultivars and treatments. Domes and reservoirs were randomly arranged within a greenhouse bench.

Cuttings were harvested at 14 and 21 days after propagation. A randomized selection of half of each treatment and cultivar were collected per harvest date. Each plant was separated by root and shoot (5 cm above the stem bottom) to measure above and below-ground dry biomass, height, stem thickness at 5.08 cm (2 in) from stem bottom, and root quality score (1–10). Successfully rooted cuttings were assigned to a classification based on the degree of adventitious rooting; a root quality score of 1–10 was assigned based on visual representation (Figure 15a–c). Propagules at day 14 were removed from their rockwool and horticultural foam media to better evaluate root quality score. Propagules at day 21 retained their treatment media, as root growth prevented separation. The effect of the media at day 21 was accounted for by subtracting the average dry weight of a sample set of rockwool and horticultural foam from the results.

#### 4.3.2. Experiment 2: Propagation System–Transplant

A transplant experiment was conducted to evaluate the effect of the propagation system (aeroponics, horticultural foam, and rockwool) and timing on transplant success, through two replicated trials. All conditions were the same as in Experiment 1, except that the aeroponic units were standardized to spray 1 min on: 1 min off timed intervals. At 8, 10, 12, and 14 days after propagating, 8 propagules from each cultivar and propagation system were randomly selected and transplanted into 4 inch pots filled with Lambert LM-111 All Purpose Mix potting media, totaling 48 propagules per transplant date. Rooted cuttings from all treatments were handled uniformly and with care during transfer. The transplants were maintained with Jack’s Professional LX All Purpose [EC of 2.1 dS·m^−1^] once daily.

Plants were removed from pots at 21 days after propagation to assess height and root quality score. Successfully rooted cuttings were assigned to a classification, based on degree of adventitious rooting; a root quality score of 0–5 was assigned based on visual representation (Figure 16).

#### 4.3.3. Experiment 3: Aeroponics–Spray Interval

Aeroponic spray timing was investigated to understand the impact of continuous and intermittent pump spray interval timings on the rate and success of root initiation and development. All aeroponic conditions were the same as in Experiment 1, except that four aeroponic systems were utilized with differentiating pump timing settings, which were compared across three trials. Trial 1 incorporated three aeroponics systems with continuous, 1 min on and 3 min off (1:3), and 1:9 timed intervals and consisting of a total of 192 cuttings (64 per treatment). Trial 2 incorporated two aeroponic systems, a continuous and 1:1 spray intervals, with 128 cuttings total. Trial 3 incorporated four aeroponic systems with continuous, 1:1, 1:3, and 1:9 spray intervals, with 256 cuttings. Each aeroponic treatment contained 32 replicates of each cultivar, totaling 64 cuttings per reservoir. Reservoirs were randomly arranged atop a greenhouse bench. Data were collected as in Experiment 1.

#### 4.3.4. Experiment 4: Aeroponics–Fertigation Dilutions

To assess how nutrient solution strengths in the aeroponics reservoir impact rooting and growth, two replicated trials were conducted, in which the electrical conductivity (EC) of the solution varied across three aeroponic systems with two replications. All aeroponic conditions were the same as in Experiment 1, except that three aeroponic systems were used with various fertigation dilutions, which were compared across two trials. Each trial utilized an aeroponic system set to spray continuously with nutrient solutions at one of three EC concentrations; initially measured to an EC of 0.7 dS·m^−1^ (equivalent to a 1:4 fertigation dilution), 1.0 dS·m^−1^ (1:3), and 1.4 dS·m^−1^ (1:2). Aeroponics systems contained 32 replicates of each cultivar with 64 cuttings per reservoir and 192 cuttings per trial. Data were collected as in Experiment 1.

#### 4.3.5. Statistical Analysis

Data analysis was conducted using R statistical software (version 2023.03.0+386) [42]. Stem diameter was taken into account in mass measurements by dividing both above and below-ground masses by stem diameter. The analysis employed mixed-effects models through the ‘lme4’ and ‘lmerTest’ packages [43,44]. In cases of count data, a Poisson distribution was utilized. The models included fixed effects for treatment, sampling day, and cultivar, along with their interactions. To account for trial-specific variability, trial was included as a random effect in all models. The ‘Anova’ function from the ‘car’ package was used for significance testing [45], employing a type II Wald Chi-squared test. Post hoc comparisons were conducted via the ‘emmeans’ package, applying Tukey’s HSD test for pairwise comparisons [46].

## 5. Conclusions

A series of experiments was conducted comparing propagation media and methods on propagation and the subsequent establishment of *Cannabis* shoot tip propagules. Further, the research has been analyzed to optimize the frequency of aeroponic spray timing and aeroponic nutrient strength. Transplant success was influenced by propagation system choice and cultivar selection, with aeroponics showing the greatest effect in enhancing transplant outcomes versus horticultural foam or rockwool. In aeroponic systems, it was identified that the use of continuous spraying obtained a maximal plant root initiation and overall growth. Optimized electrical conductivity (EC) ratios proved to positively impact root development and height. Aeroponics is advisable in environments where control and rapid root-and-shoot growth are priorities. While acknowledging its superiority in enhancing transplant outcomes, careful consideration is required to mitigate power reliance and demands, in addition to prioritizing resource conservation and sustainable agricultural practices. By considering the most suitable and effective propagation systems and, in the case of aeroponics, spray time intervals and fertigation ratios, cultivators can utilize these findings to elevate proficiency and precision, leading to successful productions of uniform and vigorous young plants. As the industry continues to expand and evolve, the cultivation of premium and consistent *Cannabis* products will be paramount. These experiments and their findings contribute to an expanding and robust knowledge foundation for future agriculture and horticulture practices, and more specifically, within the realm of *Cannabis* cultivation, reaffirming the industry’s commitment to quality control, sustainability, growth, and success.

## Figures and Tables

**Figure 1 plants-13-01256-f001:**
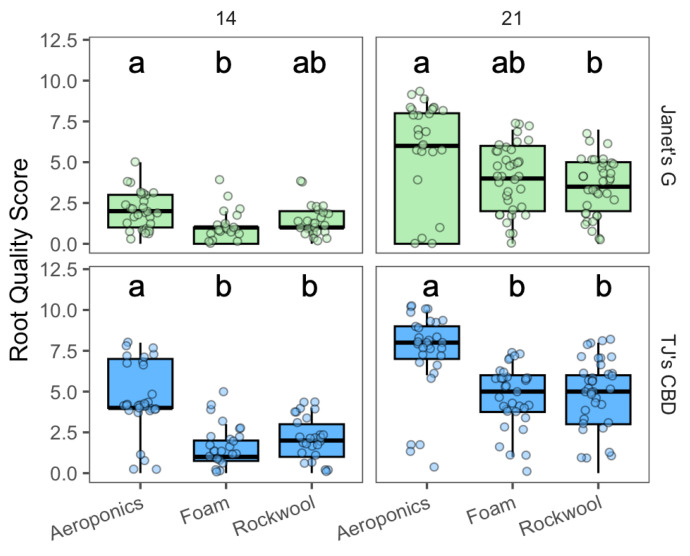
Root quality score compared across propagation systems—aeroponics, rockwool, and (horticultural) foam—using *Cannabis sativa* L. on day 14 (**left** panels) and 21 (**right** panels). Mean separation across propagation systems is indicated with letters, using Tukey’s Honest Significant Difference at *p* < 0.05.

**Figure 2 plants-13-01256-f002:**
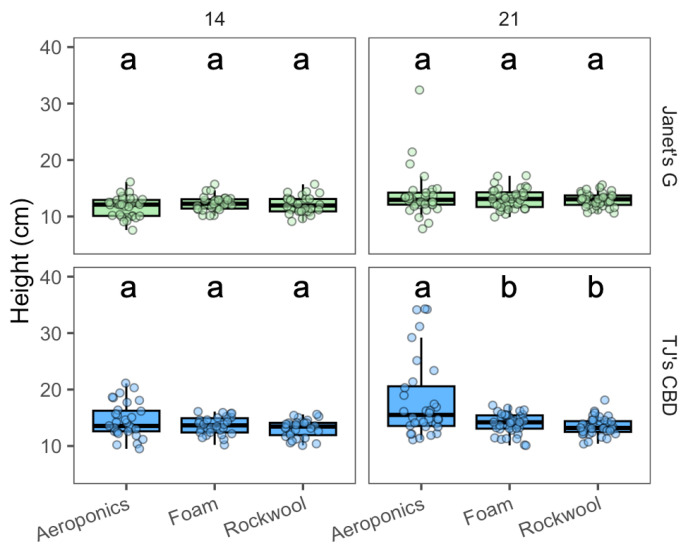
Height (cm) was compared across propagation systems—aeroponics, rockwool, and (Horticultural) foam—using *Cannabis sativa* L. Mean separation across propagation systems is indicated with letters, using Tukey’s Honest Significant Difference at *p* < 0.05.

**Figure 3 plants-13-01256-f003:**
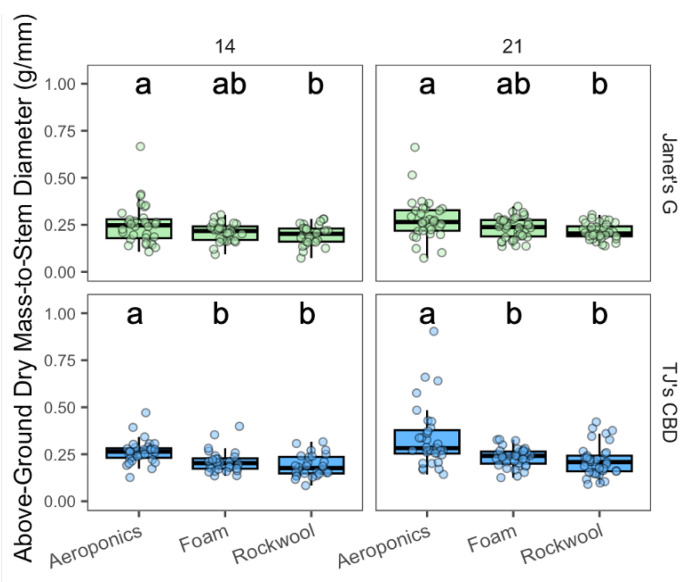
Above-ground biomass-to-stem diameter (g/mm) was compared across propagation systems—aeroponics, rockwool, and (horticultural) foam—using *Cannabis sativa* L. Stem diameter was taken into account in mass measurements by dividing above-ground masses by stem diameter. Mean separation across propagation systems is indicated with letters, using Tukey’s Honest Significant Difference at *p* < 0.05.

**Figure 4 plants-13-01256-f004:**
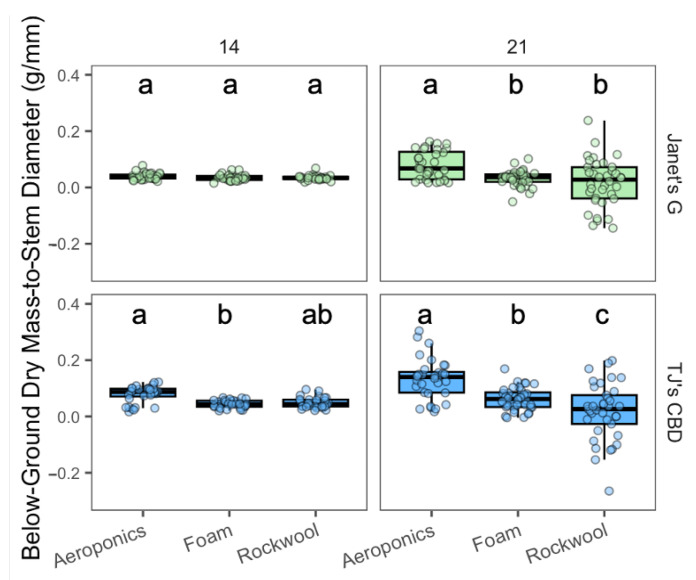
Below-ground biomass-to-stem diameter (g/mm) was compared across propagation systems—aeroponics, rockwool, and (Horticultural) foam—using *Cannabis sativa* L. Stem diameter was taken into account in mass measurements by dividing below-ground masses by stem diameter. To refine the measurement of biomass, the average weights of dry test samples—0.77 g from foam and 1.49 g from rockwool—were subtracted from the total dry masses on day 21, where media could not be separated from the roots. This adjustment allows for a more accurate comparison of the actual plant biomass across the different propagation systems. Mean separation across propagation systems is indicated with letters, using Tukey’s Honest Significant Difference at *p* < 0.05.

**Figure 5 plants-13-01256-f005:**
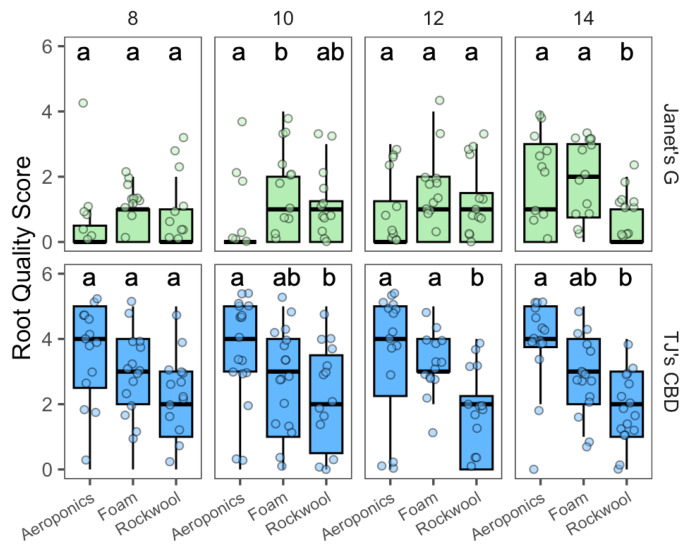
Root quality score compared across transplanted days and propagation systems—aeroponics, rockwool, and (Horticultural) foam—using *Cannabis sativa* L. Mean separation across propagation systems is indicated with letters, using Tukey’s Honest Significant Difference at *p* < 0.05.

**Figure 6 plants-13-01256-f006:**
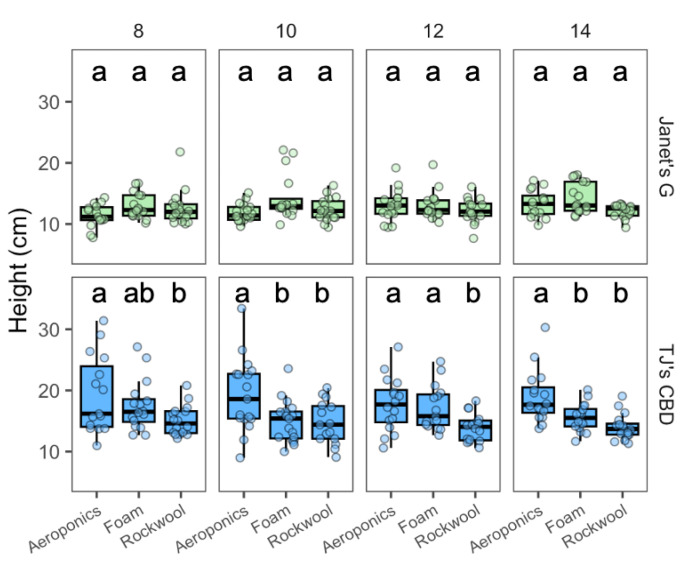
Height (cm) compared across transplanted days and propagation systems—aeroponics, rockwool, and (Horticultural) foam—using *Cannabis sativa* L. Mean separation across propagation systems is indicated with letters, using Tukey’s Honest Significant Difference at *p* < 0.05.

**Figure 7 plants-13-01256-f007:**
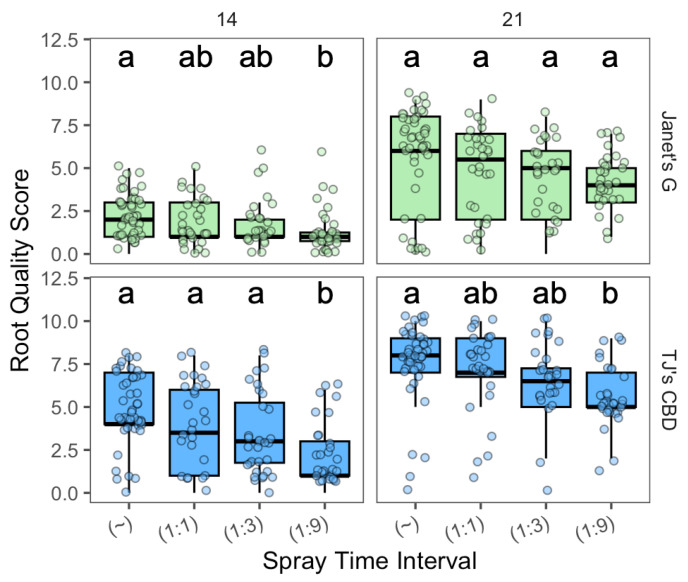
Root quality score compared across spray time intervals (min) in aeroponics systems using *Cannabis sativa* L. (~ was continuously on; 1:1, 1:3, and 1:9 were 1 min on with 1, 3, and 9 min off, respectively). Mean separation across spray time intervals is indicated with letters, using Tukey’s Honest Significant Difference at *p* < 0.05.

**Figure 8 plants-13-01256-f008:**
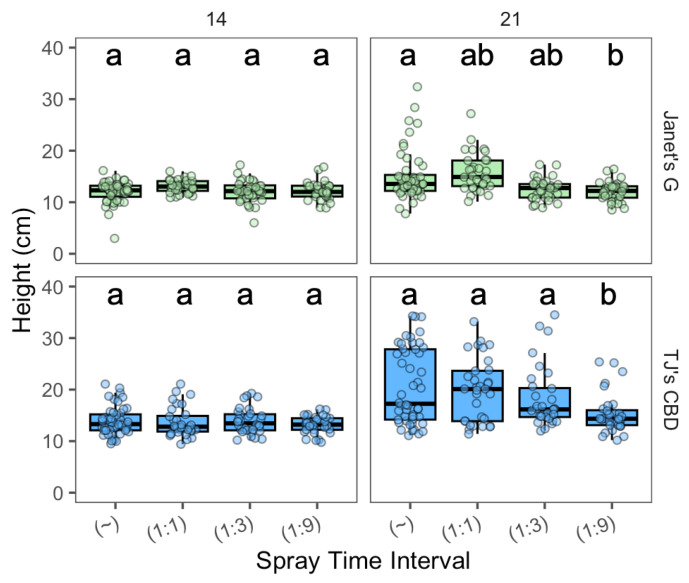
Height (cm) compared across spray time intervals (min) in aeroponic systems using *Cannabis sativa* L. (~ was continuously on; 1:1, 1:3, and 1:9 were 1 min on with 1, 3, and 9 min off, respectively). Mean separation across spray time intervals is indicated with letters, using Tukey’s Honest Significant Difference at *p* < 0.05.

**Figure 9 plants-13-01256-f009:**
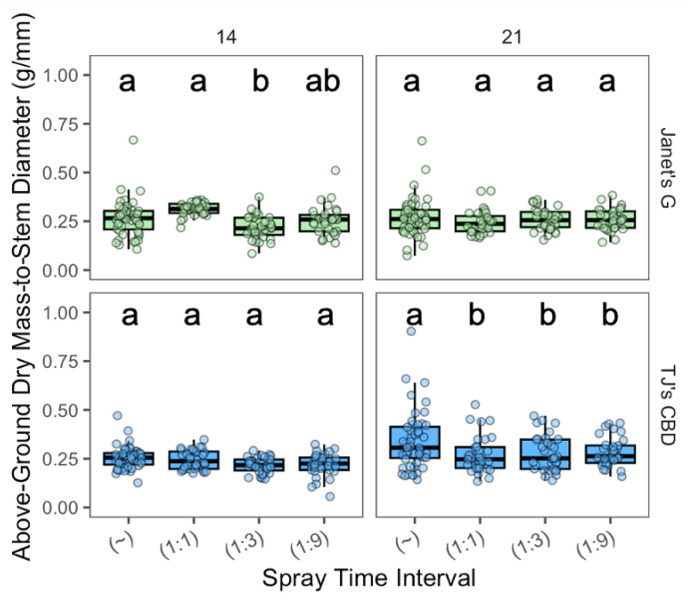
Above-ground biomass-to-stem diameter (g/mm) was compared across spray time intervals in aeroponic systems using *Cannabis sativa* L. (~ was continuously on; 1:1, 1:3, and 1:9 were 1 min on with 1, 3, and 9 min off, respectively). Stem diameter was taken into account in mass measurements by dividing above-ground mass by stem diameter. Mean separation across spray time intervals is indicated with letters, using Tukey’s Honest Significant Difference at *p* < 0.05.

**Figure 10 plants-13-01256-f010:**
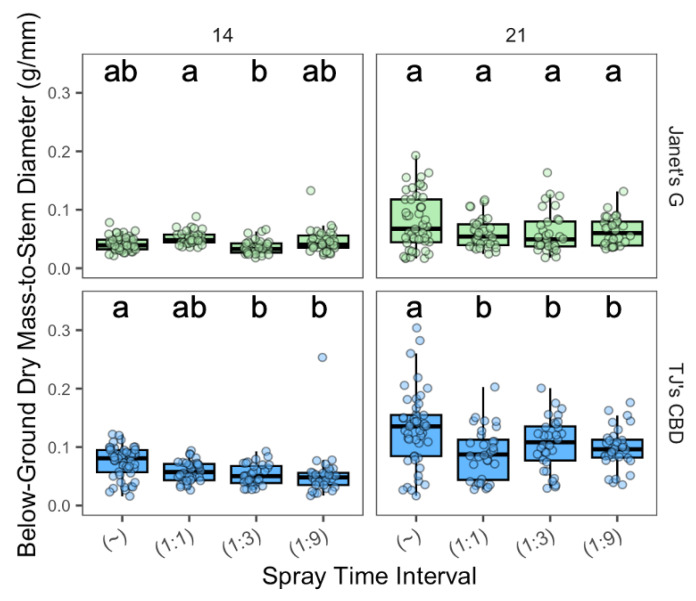
Below-ground biomass-to-stem diameter (g/mm) was compared across spray time intervals in aeroponic systems using *Cannabis sativa* L. (~ was continuously on; 1:1, 1:3, and 1:9 were 1 min on with 1, 3, and 9 min off, respectively). Stem diameter was taken into account in mass measurements by dividing below-ground masses by stem diameter. Mean separation across spray time intervals is indicated with letters, using Tukey’s Honest Significant Difference at *p* < 0.05.

**Figure 11 plants-13-01256-f011:**
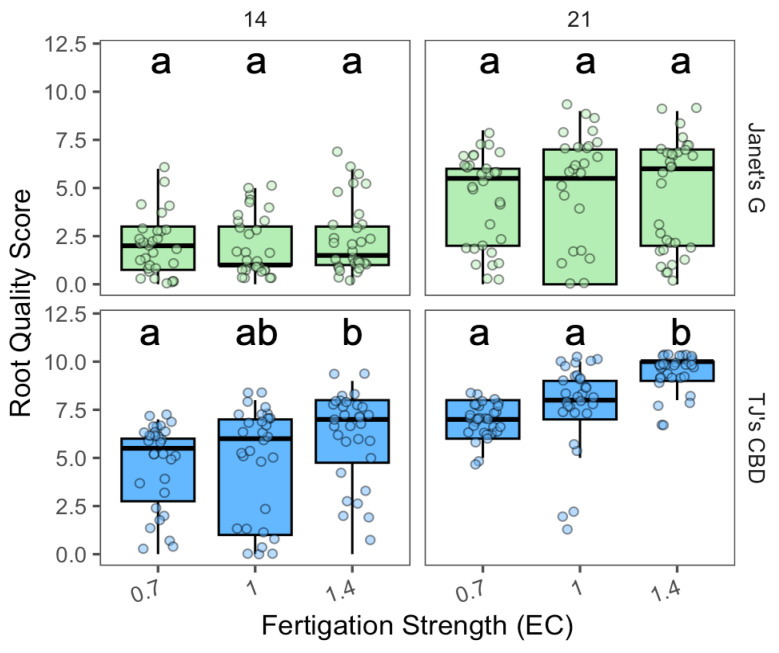
Root quality score compared across nutrient concentrations (EC in dS·m^−1^) in aeroponic systems using *Cannabis sativa* L. Mean separation across nutrient concentrations (EC) is indicated with letters, using Tukey’s Honest Significant Difference at *p* < 0.05.

**Figure 12 plants-13-01256-f012:**
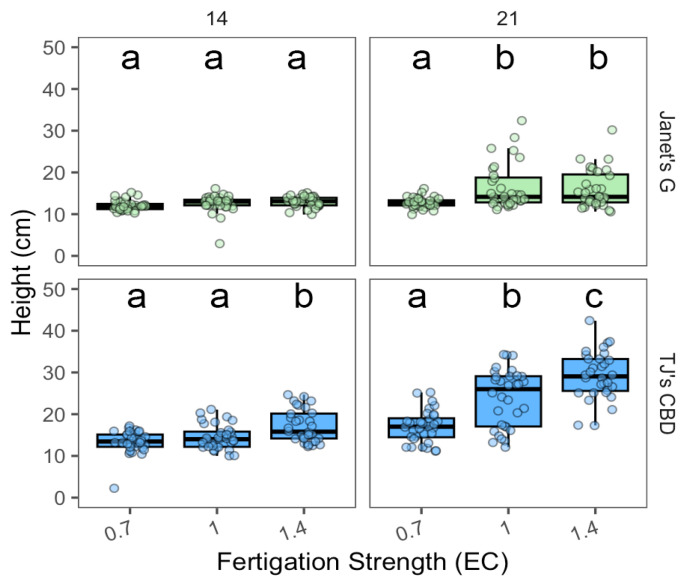
Height (cm) compared across nutrient concentrations (EC in dS·m^−1^) in aeroponic systems using *Cannabis sativa* L. Mean separation across nutrient concentrations (EC) is indicated with letters, using Tukey’s Honest Significant Difference at *p* < 0.05.

**Figure 13 plants-13-01256-f013:**
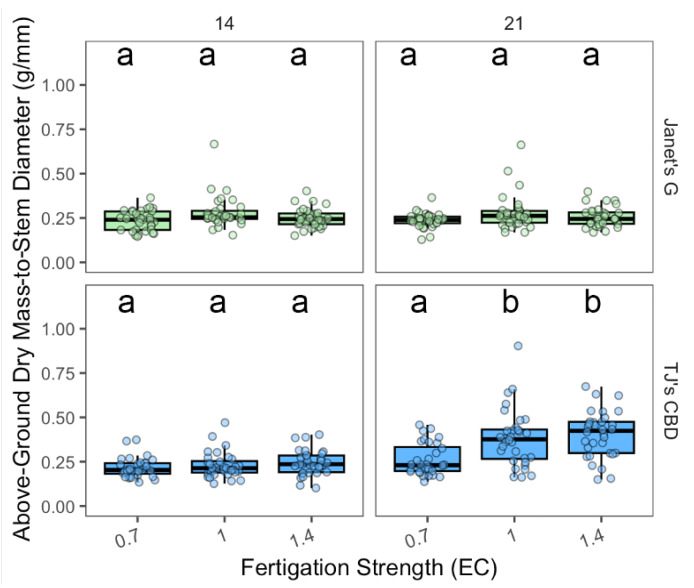
Above-ground biomass-to-stem diameter (g/mm) was compared across nutrient concentrations (EC in dS·m^−1^) in aeroponic systems using *Cannabis sativa* L. Stem diameter was taken into account in mass measurements by dividing both above-ground masses by stem diameter. Mean separation across nutrient concentration (EC) is indicated with letters, using Tukey’s Honest Significant Difference at *p* < 0.05.

**Figure 14 plants-13-01256-f014:**
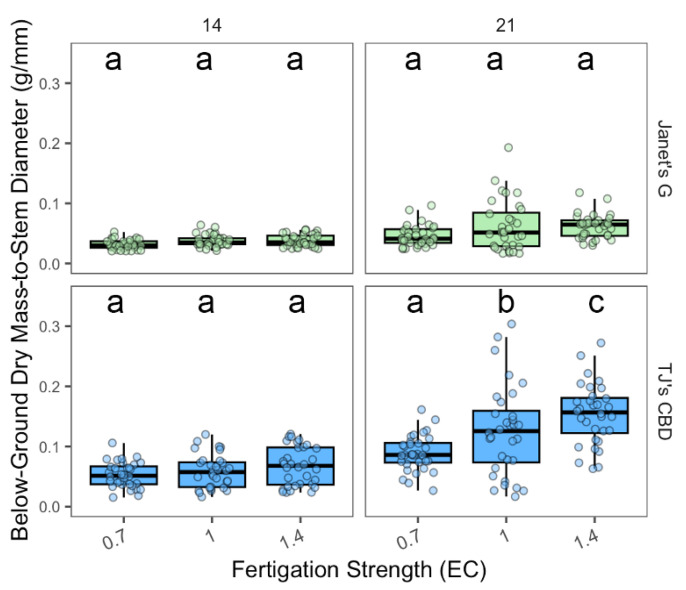
Below-ground biomass-to-stem diameter (g/mm) was compared across nutrient concentrations (EC in dS·m^−1^) in aeroponic systems using *Cannabis sativa* L. Stem diameter was taken into account in mass measurements by dividing below-ground masses by stem diameter. Mean separation across nutrient concentration (EC) is indicated with letters, using Tukey’s Honest Significant Difference at *p* < 0.05.

**Figure 15 plants-13-01256-f015:**
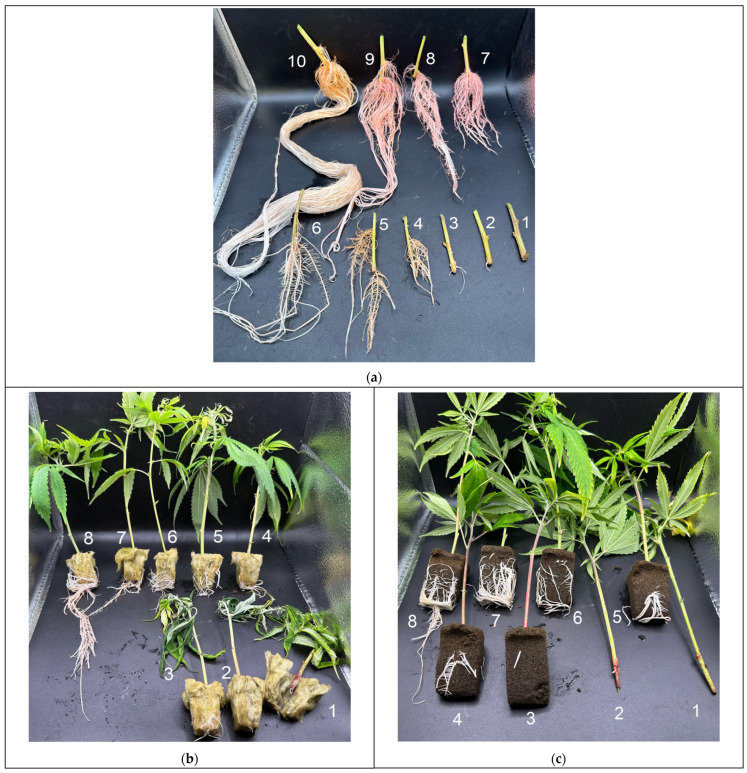
Representative photos demonstrating root quality scoring (regardless of cultivar) for aeroponics (**a**) 1–10; rockwool cubes 1–8 (**b**); horticultural foam 1–8 (**c**).

**Figure 16 plants-13-01256-f016:**
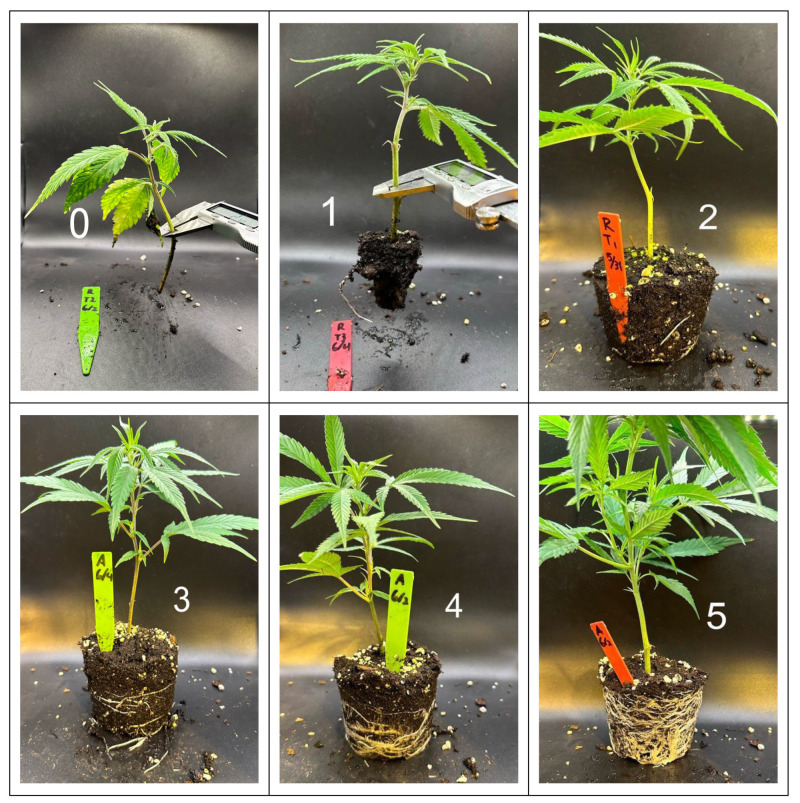
Representative photos demonstrating root quality scoring (regardless of cultivar and treatment) of Experiment 2′s out-of-pot transplanted root scores 0–5.

**Table 1 plants-13-01256-t001:** This table outlines the four experiments conducted on *Cannabis sativa* L. cuttings and the plant growth metrics measured to identify best practice strategies for propagation.

Experiment		Objective	Variables	Data Points
1	Propagation System	Comparing the efficacy of various propagation systems	Propagation System: Aeroponics, Rockwool, Horticultural Foam Cultivars: ‘TJ’s CBD’ and ‘Janet’s G’ CBG	Root Quality Score, Height (cm), Above-Ground Biomass, Below-Ground Biomass
2	Propagation System– Transplant	Assessing the impact of transplant timing and effects across various propagation systems	Transplant Days: 8, 10, 12, 14 Propagation System: Aeroponics, Rockwool, Horticultural Foam Cultivars: ‘TJ’s CBD’ and ‘Janet’s G’ CBG	Root Quality Score, Height (cm)
3	Aeroponics– Spray Intervals	Examining the effect of aeroponic spray intervals	Spray Intervals (min): Continuous, 1:1, 1:3, 1:9 Cultivars: ‘TJ’s CBD’ and ‘Janet’s G’ CBG	Root Quality Score, Height (cm), Above-Ground Biomass, Below-Ground Biomass
4	Aeroponics– Fertigation Dilutions	Analyzing the influence of fertigation dilutions in aeroponics	Fertigation Dilutions: 1:2, 1:3, 1:4 Cultivars: ‘TJ’s CBD’ and ‘Janet’s G’ CBG	Root Quality Score, Height (cm), Above-Ground Biomass, Below-Ground Biomass

**Table 2 plants-13-01256-t002:** This table summarizes the four experiment’s results. Results exhibiting each experiment’s efficacy are indicated with color and a numerical value, with higher values in green demonstrating the best performance and lower values in red demonstrating the poorest performance. The strongest performing treatment per data set is highlighted in purple. NA or omitted results indicate no differences amongst variables. Cultivar 1 (green) represents ‘Janet’s G’ CBG and Cultivar 2 (blue) represents ‘TJ’s CBD’ cultivars, consistent with the Results section figure’s color scheme. Score Total (By Experiment) corresponds to the same rows, with a bolded vertical line and font assigning treatment per experiment.

Experiment		Variables	Data Point	Day	Treatment	Cultivar 1	Cultivar 2	Score (Cultivars Combined)	Score Total (By Experiment)
1	Propagation System	Propagation System, Cultivar, Sampling Day	Root Quality	14	Aeroponics	3	3	6	
					Foam	1	1	2	
					Rockwool	2	1	3	
				21	Aeroponics	3	3	6	
					Foam	2	1	3	
					Rockwool	1	1	2	
			Height	21	Aeroponics	NA	3	3	
					Foam, Rockwool		1	1	
			Above- Ground Biomass	14	Aeroponics	3	3	6	
					Foam	2	1	3	
					Rockwool	1	1	2	
				21	Aeroponics	3	3	6	
					Foam	2	1	3	
					Rockwool	1	1	2	
			Below- Ground Biomass	14	Aeroponics	NA	3	3	
					Foam		1	1	
					Rockwool		2	2	
				21	**Aeroponics**	3	3	6	**36**
					**Foam**	1	2	3	**16**
					**Rockwool**	1	1	2	**14**
2	Propagation System: Transplant	Propagation System, Cultivar, Transplant Day	Root Quality	10	Aeroponics	1	3	4	
					Foam	3	2	5	
					Rockwool	2	1	3	
				12	Aeroponics, Foam	NA	3	3	
					Rockwool		1	1	
				14	Aeroponics	3	3	6	
					Foam	3	2	5	
					Rockwool	1	1	2	
			Height	8	**Aeroponics**	NA	3	3	**25**
					**Foam**		2	2	**18**
					**Rockwool**		1	1	**10**
				10	Aeroponics		3	3	
					Foam, Rockwool		1	1	
				12	Aeroponics		3	3	
					Foam, Rockwool		1	1	
				14	Aeroponics		3	3	
					Foam, Rockwool		1	1	
3	Aeroponics—Spray Interval (minutes on:off)	Spray Time Interval, Cultivar, Sampling Day	Root Quality	14	Continuous	3	3	6	
					1:1, 1:3	2	3	5	
					1:9	1	1	2	
				21	Continuous	NA	3	3	
					1:1, 1:3		2	2	
					1:9		1	1	
			Height	21	Continuous	3	3	6	
					1:1, 1:3	2	3	5	
					1:9	1	1	2	
			Above- Ground Biomass	14	Continuous, 1:1	3	NA	3	
					1:3	1		1	
					1:9	2		2	
				21	Continuous	NA	3	3	
					1:1,1:3,1:9		1	1	
			Below- Ground Biomass	14	**Continuous**	2	3	5	**29**
					**1:1**	3	2	5	**22**
					**1:3**	1	1	2	**17**
					**1:9**	2	1	3	**12**
				21	Continuous	NA	3	3	
					1:1,1:3,1:9		1	1	
4	Aeroponics—Fertigation Dilution (EC)	Fertigation Dilution, Cultivar, Sampling Day	Root Quality	14	0.7	NA	1	1	
					1		2	2	
					1.4		3	3	
				21	0.7, 1		1	1	
					1.4		3	3	
			Height	14	0.7, 1		1	1	
					1.4		3	3	
				21	0.7	1	1	2	
					1	3	2	5	
					1.4	3	3	6	
			Above- Ground Biomass	21	0.7	NA	1	1	
					1, 1.4		3	3	
			Below- Ground Biomass	21	**0.7**		1	1	**7**
					**1**		2	2	**14**
					**1.4**		3	3	**21**

## Data Availability

The data presented in this study are available in the Appendix A spreadsheet.

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
