# Peer review of "Evaluating Propagation Techniques for Cannabis sativa L. Cultivation: A Comparative Analysis of Soilless Methods and Aeroponic Parameters"

_plants, 2024, doi:10.3390/plants13091256_

Round 1
Reviewer 1 Report
Comments and Suggestions for Authors
Dear authors.
Performed work is valuable.
The experimental part is complicated and have to be described more clearly (maybe providing a schematic view).
I would recommend to combine the results and discussion parts together for more comprehensive analysis.
Author Response
Dear Reviewer,
Thank you for taking the time to review, all your feedback is very appreciated.
-
- Performed work is valuable.
- Many thanks for the positive feedback.
- The experimental part is complicated and has to be described more clearly (maybe providing a schematic view).
- Similarly to Results in Ratings
- Agreed. We added table 1 and 2 as an overall description of the 4 experiments and results.
- I would recommend combining the results and discussion parts together for more comprehensive analysis.
- That’s a very valid point that was well considered, though for structural purposes we’ve decided to keep separated.
Reviewer 2 Report
Comments and Suggestions for Authors
The manuscript by M. Weingarten, N. Mattson and H. Grabb evaluated propagation techniques for Cannabis sativa L. by comparing soilless methods using three propagation systems: aeroponics, rockwool, and (horticultural) foam to determine their impact on root development, plant height, above-ground dry mass of C. sativa L. As aeroponics was found to be in most cases more effective than soilless propagation substrates for root development and plant growth and root dry mass, two Cannabis cultivars were tested further using three other parameters i.e., 1) transplant timings (experiment 2); 2) aeroponic spray intervals (Experiment 3); and (3) aeroponic reservoir nutrient concentrations (experiment 4).
The authors concluded from their study “compelling evidence that aeroponics can yield equal or superior root development, plant growth and transplant success” and that “the use of continuous spray intervals and optimized nutrient concentrations which promotes root and overall plant growth in aeroponics.”
Here are some suggestions
Figure 16- images could be in one row or is there another image required to complete the set? Please state which cultivar was used for figure 16. Was there a difference with two cultivars? Are images available for both cultivars?
The same for Fig. 15, please state which cultivar in legend.
It would be interesting to know- was there a difference in the level of cannabidiol and cannabigerol in plants grown in different aeroponics conditions? Could that be discussed or is there any data available?
Author Response
Dear Reviewer,
- Thank you for taking the time to review, all your feedback is very appreciated.
-
- The authors concluded from their study “compelling evidence that aeroponics can yield equal or superior root development, plant growth and transplant success” and that “the use of continuous spray intervals and optimized nutrient concentrations which promotes root and overall plant growth in aeroponics.”
- Here are some suggestions
- Figure 16- images could be in one row or is there another image required to complete the set? Please state which cultivar was used for figure 16. Was there a difference with two cultivars? Are images available for both cultivars?
- The same for Fig. 15, please state which cultivar in legend.
- The rooting scale was the same for both cultivars (and the reference photos were used for evaluating both), we added a note to the figure legend pointing this out. Cultivars are NA, these are rooting scales.
- For Figure 16, Agreed, corrected to complete the row.
- It would be interesting to know- was there a difference in the level of cannabidiol and cannabigerol in plants grown in different aeroponics conditions? Could that be discussed or is there any data available?
- Although an interesting topic, the focus of this project was on propagation and the early transplant so unfortunately we did not have the scope to grow the plants out to the final flowering stage.
Reviewer 3 Report
Comments and Suggestions for Authors
The work is certainly of interest, given the increasing demand for hemp propagation material for cultivations of low-THC cultivars.
Several aspects of the text need to be reviewed to bring it in line with the high standard of the journal and to better highlight the results obtained.
- Affiliations: Avoid repeating the same institution and address multiple times.
- Abstract: Provides a clear summary of the text.
- Keywords: Is it necessary for them to be numbered? Consider removing numbering.
- The bibliographic references in the text need to be standardized to match the journal's standard.
- Introduction: Provides a clear overview of the current state of the art, though minor typos need correction.
- Units of measurement: Ensure adherence to the International System of Units.
- Methodology: The description is a bit confusing; perhaps it would be better to outline it in tables. Justification for the use of rock wool should be provided due to sustainability concerns related to waste management.
- Results: Quite well-presented and clear, particularly aided by the included graphs. Consider exploring combinations of investigated factors.
- Discussion: Compare obtained results with relevant literature. Given the experimental design's uniqueness, it would be better integrate it point by point within the results.
- Acknowledgments: Limit acknowledgments to the scientific aspect; include information about research funding, if applicable.
- In summary, the work is promising but requires careful revision before acceptance.
Comments on the Quality of English LanguageThe English is of an excellent standard; I recommend a proofreading for minor typographical errors.
Author Response
Dear Reviewer,
- Thank you for taking the time to review, all your feedback is very appreciated.
-
- The work is certainly of interest, given the increasing demand for hemp propagation material for cultivations of low-THC cultivars.
- Several aspects of the text need to be reviewed to bring it in line with the high standard of the journal and to better highlight the results obtained.
- Affiliations: Avoid repeating the same institution and address multiple times.
- Agreed, corrected.
- Abstract: Provides a clear summary of the text.
- Keywords: Is it necessary for them to be numbered? Consider removing numbering.
- Agreed, corrected.
- The bibliographic references in the text need to be standardized to match the journal's standard.
- Agreed, Corrected
- Introduction: Provides a clear overview of the current state of the art, though minor typos need correction.
- Corrections in first sentence and last two intro paragraphs.
- Units of measurement: Ensure adherence to the International System of Units.
- [check units of measurement - convert to metric]
- Thanks for catching that, we found two instances where inches were mentioned and these have been corrected to cm.
- Methodology: The description is a bit confusing; perhaps it would be better to outline it in tables.
- Agreed, as noted for reviewer 1 we added Table 1 summarizing the 4 experiments.
- Justification for the use of rock wool should be provided due to sustainability concerns related to waste management.
- Despite its sustainability concerns (which we agree with) rockwool remains the standard commercial substrate for propagation of hemp cuttings, therefore we included it as a comparison, this was noted in the introduction and discussion.
- Results: Quite well-presented and clear, particularly aided by the included graphs. Consider exploring combinations of investigated factors.
- We understand your point regarding exploring combination factors. However in our experiments, within a cultivar we only looked at a single treatment factor at a time. Future research should look at interacting factors (such as fertilizer concentration and mist interval).
- Discussion: Compare obtained results with relevant literature. Given the experimental design's uniqueness, it would be better to integrate it point by point within the results.
- The reviewer has a valid point. Unfortunately, however, there is very little existing literature on hemp propagation and we cited the few studies we could find. We expanded our search to look at other crops beyond hemp
- Acknowledgments: Limit acknowledgments to the scientific aspect; include information about research funding, if applicable.
- We shortened the acknowledgements text. There was no specific funding source, the project was executed as part of a student capstone project.
- In summary, the work is promising but requires careful revision before acceptance.
- Many thanks for the feedback.
Reviewer 4 Report
Comments and Suggestions for Authors
I think this manuscript should make some modifications for publication in this journal for the following reasons:
1、In experiment 1, in general, the development of the root system of a plant is directly proportional to the nutritional status of the above-ground part of the plant, but the results of this experiment and the discussion didn’t mention the relationship between root quality score and biomass to stem diameter below ground and above ground, suggesting that this part of the discussion should be added, or that the impact of aeroponics on the rooting of cannabis and growth of cannabis plugs should be emphasized with regard to the use of resources.
2、In experiment 2, the results showed that there was almost no improvement in the root quality score of Janet's G under aeroponic conditions from the 8th day to the 14th day, whether this was due to damage to the root system during transplanting? Besides, are horticultural foam and rockwool cleaned when transplanted, and if so how do you ensure that there is no damage to the root system, and if not how do you rule out the effects of the substrate on the growth of the plugs?
3、In experiment 3, is there a theoretical basis for choices of spray interval? Please clarify or cite the reasons for setting 1:1, 1:3 and 1:9.

Comments on the Quality of English LanguageFairly well presented in English.
Author Response
Dear Reviewer,
- Thank you for taking the time to review, all your feedback is very appreciated.
- I think this manuscript should make some modifications for publication in this journal for the following reasons:
- In experiment 1, in general, the development of the root system of a plant is directly proportional to the nutritional status of the above-ground part of the plant, but the results of this experiment and the discussion didn’t mention the relationship between root quality score and biomass to stem diameter below ground and above ground, suggesting that this part of the discussion should be added, or that the impact of aeroponics on the rooting of cannabis and growth of cannabis plugs should be emphasized with regard to the use of resources.
- This general trend was displayed in each experiment for all treatments, we added a sentence in the discussion section noting this general trend in addition to Table 2.
- In experiment 2, the results showed that there was almost no improvement in the root quality score of Janet's G under aeroponic conditions from the 8th day to the 14th day, whether this was due to damage to the root system during transplanting? Besides, are horticultural foam and rockwool cleaned when transplanted, and if so how do you ensure that there is no damage to the root system, and if not how do you rule out the effects of the substrate on the growth of the plugs?
- We don’t believe there was major root damage at the transplanting stage, we treated the rooted cuttings similar to commercial industry members in terms of transplanting. Rooted cuttings from all treatments were handled uniformly/carefully. A sentence was added to the methods detailing the transplanting method
- In experiment 3, is there a theoretical basis for choices of spray interval? Please clarify or cite the reasons for setting 1:1, 1:3 and 1:9.
- We decided to use increments which would encompass a wide range in variation of timer settings based on our observations of commercial facilities (for example, if the 1:9 was successful it would have had a 90% lower electricity use for the pump than continuous spray).
Round 2
Reviewer 2 Report
Comments and Suggestions for Authors
The manuscript had improved due by the inclusion of tables and helps the reader.
A few comments.
For Figure 15(a-c) it is important to mention which cultivars were used to get this figure in the legend. It is missing in the legend. Are we assuming that both cultivars looked the same?
line 601-602 Please modify this sentence or title of figure 16 ; 'Figure 16. Demonstrates the root quality score of Experiment 2’s out-of-pot transplanted root scores 0-5. no mention of cultivar.
Author Response
The manuscript had improved due by the inclusion of tables and helps the reader.
A few comments.
For Figure 15(a-c) it is important to mention which cultivars were used to get this figure in the legend. It is missing in the legend. Are we assuming that both cultivars looked the same?
- Although a good point, we are looking at root formation visual density, regardless of cultivar used, for universal utilization. Updated the figure legend to elaborate more thoroughly.
line 601-602 Please modify this sentence or title of figure 16 ; 'Figure 16. Demonstrates the root quality score of Experiment 2’s out-of-pot transplanted root scores 0-5. no mention of cultivar.
- Same as previous comment.
Thank you for taking the time to review and provide feedback!
Reviewer 3 Report
Comments and Suggestions for Authors
Compared to the previous version, the entire work appears more organized. Data, results, and conclusions are presented clearly, highlighting their complexity in the most appropriate way.
Author Response
Compared to the previous version, the entire work appears more organized. Data, results, and conclusions are presented clearly, highlighting their complexity in the most appropriate way.
- Thank you for the taking the time to review and provide feedback!
Reviewer 4 Report
Comments and Suggestions for Authors
The manuscript entitled “Evaluating Propagation Techniques for Cannabis sativa L. Cultivation: a Comparative Analysis of Soilless Methods and Aeroponic Parameters” is based on a multifaceted experiment with Cannabis as experimental material. Cannabis is a multifaceted experiment that evaluates propagation techniques for Cannabis sativa L. nutritional plugs for growth and developmental proficiency, with important implications for the realization of high quality Cannabis harvests and resource conservation. However, the manuscript also suffers from many fundamental errors, redundant sections and incomplete representations. Please refer to the following comments for revisions.
1. In Experiment 3, the choice of spray intervals was, in my opinion, not sufficiently scientifically justified. In addition, in the time interval treatment of 1:1-1:9, it shows a tendency of increasing first and then decreasing, how to ensure that the time interval between 1:3 and 1:9 must be lower than the effect of continuous spraying?
2. Please improve the clarity of the pictures in the manuscripts.
3. The depth of the discussion part is not enough, it is suggested to increase the number of references.
4. The pictures in figure15 are too cluttered, please optimize.
Comments on the Quality of English Language
The overall English quality of the manuscript is good.
Author Response
- In Experiment 3, the choice of spray intervals was, in my opinion, not sufficiently scientifically justified. In addition, in the time interval treatment of 1:1-1:9, it shows a tendency of increasing first and then decreasing, how to ensure that the time interval between 1:3 and 1:9 must be lower than the effect of continuous spraying?
- Created further justification with references that hopefully satifies. The idea here was based on inquiries from commercial growers interested to save on energy costs and resource preservation by reducing the amount of time the spray pump was on (i.e. motivated by industry application rather than science), additionally it was found that different species prefer different timings.
- I’m not sure where this observation was found or I may be misunderstanding. Can you please indicate specifically where or elaborate?
- Please improve the clarity of the pictures in the manuscripts.
- Agreed, updated by organizing loose figures and improved figure legends.
- The depth of the discussion part is not enough, it is suggested to increase the number of references.
- Agreed, updated discussion to be a bit more impactful.
- The pictures in figure 15 are too cluttered, please optimize.
- Agreed, updated by organizing in a table
Thank you for taking the time to review and provide feedback!
Round 3
Reviewer 2 Report
Comments and Suggestions for Authors
The authors addressed all my comments to improve the manuscript.
Author Response
Thank you for taking the time to review!